# Landmark Ordinal Embedding

**Nikhil Ghosh**[*]
UC Berkeley
nikhil_ghosh@berkeley.edu

**Yuxin Chen**[*]
UChicago
chenyuxin@uchicago.edu

**Yisong Yue**
Caltech
yyue@caltech.edu

## Abstract

In this paper, we aim to learn a low-dimensional Euclidean representation from a set of constraints of the form "item $j$ is closer to item $i$ than item $k$". Existing approaches for this "ordinal embedding" problem require expensive optimization procedures, which cannot scale to handle increasingly larger datasets. To address this issue, we propose a landmark-based strategy, which we call *Landmark Ordinal Embedding* (LOE). Our approach trades off statistical efficiency for computational efficiency by exploiting the low-dimensionality of the latent embedding. We derive bounds establishing the statistical consistency of LOE under the popular Bradley-Terry-Luce noise model. Through a rigorous analysis of the computational complexity, we show that LOE is significantly more efficient than conventional ordinal embedding approaches as the number of items grows. We validate these characterizations empirically on both synthetic and real datasets. We also present a practical approach that achieves the "best of both worlds", by using LOE to warm-start existing methods that are more statistically efficient but computationally expensive.

## 1 Introduction

Understanding similarities between data points is critical for numerous machine learning problems such as clustering and information retrieval. However, we usually do not have a "good" notion of similarity for our data *a priori*. For example, we may have a collection of images of objects, but the natural Euclidean distance between the vectors of pixel values does not capture an interesting notion of similarity. To obtain a more natural similarity measure, we can rely on (weak) supervision from an oracle (e.g. a human expert). A popular form of weak supervision is ordinal feedback of the form "item $j$ is closer to item $i$ than item $k$" [8]. Such feedback has been shown to be substantially more reliable to elicit than cardinal feedback (i.e. how close item $i$ is to item $j$), especially when the feedback is subjective [11]. Furthermore, ordinal feedback arises in a broad range of real-world domains, most notably in user interaction logs from digital systems such as search engines and recommender systems [10, 15, 2, 24, 19, 17].

We thus study the ordinal embedding problem [18, 22, 14, 13, 21], which pertains finding low-dimensional representations that respect ordinal feedback. One major limitation with current state-of-the-art ordinal embedding methods is their high computational complexity, which often makes them unsuitable for large datasets. Given the dramatic growth in real-world dataset sizes, it is desirable to develop methods that can scale computationally.

**Our contribution.** In this paper, we develop computationally efficient methods for ordinal embedding that are also statistically consistent (i.e. run on large datasets and converge to the "true" solution with enough data). Our method draws inspiration from Landmark Multidimensional Scaling [7], which approximately embeds points given distances to a set of "landmark" points. We adapt this technique to the ordinal feedback setting, by using results from ranking with pairwise comparisons

---

[*]Research done when Nikhil Ghosh and Yuxin Chen were at Caltech.

and properties of Euclidean distance matrices. The result is a fast embedding algorithm, which we call Landmark Ordinal Embedding (LOE).

We provide a thorough analysis of our algorithm, in terms of both the sample complexity and the computational complexity. We prove that LOE recovers the true latent embedding under the Bradley-Terry-Luce noise model [4, 16] with a sufficient number of data samples. The computational complexity of LOE scales linearly with respect to the number of items, which in the large data regime is orders of magnitude more efficient than conventional ordinal embedding approaches. We empirically validate these characterizations on both synthetic and real triplet comparison datasets, and demonstrate dramatically improved computational efficiency over state-of-the-art baselines.

One trade-off from using LOE is that it is statistically less efficient than previous approaches (i.e. needs more data to precisely characterize the "true" solution). To offset this trade-off, we empirically demonstrate a "best of both worlds" solution by using LOE to *warm-start* existing state-of-the-art embedding approaches that are statistically more efficient but computationally more expensive. We thus holistically view LOE as a first step towards analyzing the statistical and computational trade-offs of ordinal embedding in the large data regime.

## 2   Related Work

**Multidimensional Scaling**   In general, the task of assigning low-dimensional Euclidean coordinates to a set of objects such that they approximately obey a given set of (dis)similarity relations, is known as Euclidean embedding. When given an input matrix $D$ of pairwise dissimilarities, finding an embedding with inter-point distances aligning with $D$ is the classical problem of (metric) Multidimensional Scaling (MDS). This problem has a classical solution that is guaranteed to find an embedding exactly preserving $D$, if $D$ actually has a low-dimensional Euclidean structure [3]. The algorithm finds the global optimum of a sensible cost function in closed form, running in approximately $O(dn^2)$ time, where $n$ is the number of objects and $d$ is the dimension of the embedding.

**Landmark MDS**   When $n$ becomes too large the MDS algorithm may be too expensive in practice. The bottleneck in the solution is the calculation of the top $d$ eigenvalues of the $n \times n$ matrix $D$. When $n$ is very large, there exists a computationally efficient approximation known as Landmark MDS (LMDS) [7]. LMDS first selects a subset of $l$ "landmark" points, where $l \ll n$, and embeds these points using classical MDS. It then embeds the remaining points using their distances to the landmark points. This "triangulation" procedure corresponds to computing an affine map. If the affine dimension of the landmark points is at least $d$, this algorithm has the same guarantee as classical MDS in the noiseless scenario. However, it runs in time roughly $O(dln + l^3)$, which is linear in $n$. The main drawback is that it is more sensitive to noise, the sensitivity being heavily dependent on the "quality" of the chosen landmarks.

**Ordinal embedding**   Currently there exist several techniques for ordinal embedding. Generalized Non-metric Multidimensional Scaling (GNMDS) [1] takes a max-margin approach by minimizing hinge loss. Stochastic Triplet Embedding (STE) [22] assumes the Bradley-Terry-Luce (BTL) noise model [4, 16] and minimizes logistic loss. The Crowd Kernel [20] and t-STE [22] propose alternative non-convex loss measures based on probabilistic generative models. All of these approaches rely on expensive gradient or projection computations and are unsuitable for large datasets. The results in these papers are primarily empirical and focus on minimizing prediction error on unobserved triplets. Recently in Jain et al. (2016) [9], theoretical guarantees were made for recovery of the true distance matrix using the maximum likelihood estimator for the BTL model.

## 3   Problem Statement

In this section, we formally state the ordinal embedding problem. Consider $n$ objects $[n] = \{1, \dots, n\}$ with respective unknown embeddings $\mathbf{x}_1, \dots, \mathbf{x}_n \in \mathbb{R}^d$. The Euclidean distance matrix $D^*$ is defined so that $D^*_{ij} = \|\mathbf{x}_i - \mathbf{x}_j\|_2^2$. Let $\mathcal{T} := \{\langle i, j, k \rangle : 1 \le i \ne j \ne k \le n, j < k\}$ be the set of unique triplets. We have access to a noisy triplet comparison oracle $\mathcal{O}$, which when given a triplet $\langle i, j, k \rangle \in \mathcal{T}$ returns a binary label $+1$ or $-1$ indicating if $D^*_{ij} > D^*_{ik}$ or not. We assume that $\mathcal{O}$

makes comparisons as follows:

$$\mathcal{O}(\langle i, j, k \rangle) = \begin{cases} +1 & \text{w.p. } f(D_{ij}^* - D_{ik}^*) \\ -1 & \text{w.p. } 1 - f(D_{ij}^* - D_{ik}^*) \end{cases}, \tag{3.1}$$

where $f : \mathbb{R} \to [0, 1]$ is the known link function. In this paper, we consider the popular BTL model [16], where $f$ corresponds to the logistic function: $f(\theta) = \frac{1}{1+\exp(-\theta)}$. In general, the ideas in this paper can be straightforwardly generalized to any linear triplet model which says $j$ is farther from $i$ than $k$ with probability $F(D_{ij} - D_{ik})$, where $F$ is the CDF of some 0 symmetric random variable. The two most common models of this form are the BTL model, where $F$ is the logistic CDF, and the Thurstone model, where $F$ is the normal CDF [5].

Our goal is to recover the points $\mathbf{x}_1, \ldots, \mathbf{x}_n$. However since distances are preserved by orthogonal transformations, we can only hope to recover the points up to an orthogonal transformation. For this reason, the error metric we will use is the Procrustes distance [12] defined as:

$$d(\mathbf{X}, \widehat{\mathbf{X}}) := \min_{Q \in O(n)} \left\| \mathbf{X} - Q\widehat{\mathbf{X}} \right\|_F, \tag{3.2}$$

where $O(n)$ is the group of $n \times n$ orthogonal matrices.

We now state our problem formally as follows:

**Problem 1** (Ordinal Embedding). *Consider $n$ points $\mathbf{X} = [\mathbf{x}_1, \ldots, \mathbf{x}_n] \in \mathbb{R}^{d \times n}$ centered about the origin. Given access to oracle $\mathcal{O}$ in* (3.1) *and budget of $m$ oracle queries, output an embedding estimate $\widehat{\mathbf{X}} = [\widehat{\mathbf{x}}_1, \ldots, \widehat{\mathbf{x}}_n]$ minimizing the Procrustes distance $d(\mathbf{X}, \widehat{\mathbf{X}})$*

# 4 Landmark Ordinal Embedding

In this section, we present our algorithm Landmark Ordinal Embedding (LOE), for addressing the ordinal embedding problem as stated in Problem 1. Instead of directly optimizing a maximum likelihood objective, LOE recovers the embedding in two stages. First, LOE estimates $\ell = O(d)$ columns of $D^*$. Then it estimates the embedding $\mathbf{X}$ from these $\ell$ columns via LMDS. We provide the pseudocode in Algorithm 1. In the remainder of this section, we focus on describing the first stage, which is our main algorithmic contribution.

## 4.1 Preliminaries

**Notation**  We establish some notation and conventions. For vectors the default norm is the $\ell_2$-norm. For matrices, the default inner product/norm is the Fröbenius inner product/norm.

Table 1: List of Notation

| | | | |
|---:|---|---:|---|
| $\mathbb{EDM}^\ell$ | cone of $\ell \times \ell$ Euclidean distance matrices | $J$ | $\mathbf{1}_\ell \mathbf{1}_\ell^\top - I_\ell$ |
| $\mathcal{P}_C$ | projection onto closed, convex set $C$ | $J^\perp$ | $\{X \in \mathbb{R}^{\ell \times \ell} : \langle X, J \rangle = 0\}$ |
| $\text{dist}(x, C)$ | $\|x - \mathcal{P}_C(x)\|$ i.e. distance of point $x$ to $C$ | $\mathbb{S}^\ell$ | $\{X \in \mathbb{R}^{\ell \times \ell} : X^\top = X\}$ |
| $X^\dagger$ | the pseudoinverse of $X$ | $\mathcal{V}$ | $\mathbb{S}^\ell \cap J^\perp$ |
| $\binom{n}{2}$ | $\{(j, k) : 1 \le j < k \le n\}$ | $\sigma_X$ | $\langle X, J \rangle / \|J\|^2$ |

**Consistent estimation**  We will say that for some estimator $\widehat{a}$ and quantity $a$ that $\widehat{a} \approx a$ with respect to a random variable $\epsilon$ if $\|\widehat{a} - a\| \le g(\epsilon)$ for some $g : \mathbb{R} \to \mathbb{R}$ which is monotonically increasing and $g(x) \to 0$ as $x \to 0$. Note that $\approx$ is an equivalence relation. If $\widehat{a} \approx a$, we say $\widehat{a}$ approximates or estimates $a$, with the approximation improving to an equality as $\epsilon \to 0$.

**Prelude: Ranking via pairwise comparisons**  Before describing the main algorithm, we discuss some results associated to the related problem of ranking via pairwise comparisons which we will use later. We consider the parametric BTL model for binary comparisons.

Assume there are $n$ items $[n]$ of interest. We have access to a pairwise comparison oracle $\mathcal{O}$ parametrized by an unknown score vector $\theta^* \in \mathbb{R}^n$. Given $\langle j, k \rangle \in \binom{n}{2}$ the oracle $\mathcal{O}$ returns:

$$\mathcal{O}(\langle j, k \rangle) = \begin{cases} 1 & \text{w.p. } f(\theta_j^* - \theta_k^*) \\ -1 & \text{w.p. } 1 - f(\theta_j^* - \theta_k^*) \end{cases},$$

---

**Algorithm 1** Landmark Ordinal Embedding (LOE)

---

1: **Input**: # of items $n$; # of landmarks $\ell$; # of samples per column $m$; dimension $d$; triplet comparison oracle $\mathcal{O}$; regularization parameter $\lambda$;
2: Randomly select $\ell$ landmarks from $[n]$ and relabel them so that landmarks are $[\ell]$
3: **for all** $i \in [\ell]$ **do**
4:     Form comparison oracle $\mathcal{O}_i(\langle j, k \rangle)$ in Eq. (4.3)
5:     $R_i \leftarrow$ regularized MLE estimate Eq. (4.1)    ⟵ ▷ *rank landmark columns*
6: for ranking from $\mathcal{O}_i$ using $m$ comparisons
7: **end for**
8: $R \leftarrow [R_1, \ldots, R_\ell]$
9: $\widetilde{W} \leftarrow R(1 : \ell - 1, 1 : \ell)$
10: $W = \begin{cases} \widetilde{W}_{i, j - \mathbf{1}(j > i)} & i \neq j \\ 0 & i = j \end{cases} \forall i, j \in [\ell]$    ⟵ ▷ *form associated $W$*
11: $J \leftarrow \mathbf{1}_\ell \mathbf{1}_\ell^\top - I_\ell$
12: $\widehat{\sigma} \leftarrow$ least-squares solution to (4.8)
13: $\widehat{\sigma}_E \leftarrow \lambda_2(\mathcal{P}_{\mathbb{S}^\ell}(W + J \cdot \text{diag}(s)))$    ⟵ ▷ *estimate column shifts $\widehat{s}$*
14: $\widehat{s} \leftarrow \widehat{\sigma} + \widehat{\sigma}_E$
15: $\widehat{E} \leftarrow \mathcal{P}_{\mathbb{EDM}^\ell}(W + J \cdot \text{diag}(\widehat{s}))$    ⟵ ▷ *estimate first $\ell$ columns of $D^*$*
16: $\widehat{F} \leftarrow R(\ell : n - 1, 1 : \ell) + \mathbf{1}_{n-\ell} \cdot \widehat{s}^\top$
17: $\widehat{\mathbf{X}} \leftarrow \text{LMDS}(\widehat{E}, \widehat{F})$ ⟵ ▷ *estimate ordinal embedding*
18: **Output**: Embedding $\widehat{\mathbf{X}}$

---

where as before $f$ is the logistic function. We call the oracle $\mathcal{O}$ a $\theta^*$-oracle.

Given access to a $\theta^*$-oracle $\mathcal{O}$, our goal is to estimate $\theta^*$ from a set of $m$ pairwise comparisons made uniformly at random labeled, that are labeled by $\mathcal{O}$. Namely, we wish to estimate $\theta^*$ from the comparison set $\Omega = \{(j_1, k_1, \mathcal{O}(\langle j_1, k_1 \rangle)), \ldots, (j_m, k_m, \mathcal{O}(\langle j_m, k_m \rangle))\}$ where the pairs $\langle j_i, k_i \rangle$ are chosen i.i.d uniformly from $\binom{n}{2}$. We refer to this task as ranking from a $\theta^*$-oracle using $m$ comparisons. To estimate $\theta^*$, we use the $\ell_2$-regularized maximum likelihood estimator:

$$\widehat{\theta} = \arg\max_\theta \mathcal{L}_\lambda(\theta; \Omega), \tag{4.1}$$

where $\mathcal{L}_\lambda(\theta; \Omega) := \sum_{(i,j) \in \Omega_+} \log(f(\theta_i - \theta_j)) + \sum_{(i,j) \in \Omega_-} \log(1 - f(\theta_i - \theta_j)) - \frac{1}{2}\lambda \|\theta\|_2^2$ for some regularization parameter $\lambda > 0$ and $\Omega_\pm := \{(i, j) : (i, j, \pm 1) \in \Omega\}$.

Observe that $[\theta] = \{\theta + s\mathbf{1}_n : s \in \mathbb{R}\}$ forms an equivalence class of score vectors in the sense that each score vector in $[\theta]$ forms an identical comparison oracle. In order to ensure identifiability for recovery, we assume $\sum_i \theta_i^* = 0$ and enforce the constraint $\sum_i \widehat{\theta}_i = 0$. Now we state the central result about $\widehat{\theta}$ that we rely upon.

**Theorem 2** (Adapted from Theorem 6 of [6]). *Given $m = \Omega(n \log n)$ observations of the form $(j, k, \mathcal{O}(\langle j, k \rangle))$ where $\langle j, k \rangle$ are drawn uniformly at random from $\binom{n}{2}$ and $\mathcal{O}$ is a $\theta^*$-oracle, with probability exceeding $1 - O(n^{-5})$ the regularized MLE $\widehat{\theta}$ with $\lambda = \Theta(\sqrt{n^3 \log n / m})$ satisfies:*

$$\left\| \theta^* - \widehat{\theta} \right\|_\infty = O\left( \sqrt{\frac{n \log n}{m}} \right). \tag{4.2}$$

If it is not the case that $\sum_i \theta_i^* = 0$, we can still apply Theorem 2 to the equivalent score vector $\theta^* - \bar{\theta}^* \mathbf{1}$, where $\bar{\theta}^* = \frac{1}{n} \mathbf{1}_n^\top \theta^*$. In place of Eq. (4.2) this yields $\left\| \theta^* - (\widehat{\theta} + \bar{\theta}^* \mathbf{1}) \right\|_\infty = O\left( \sqrt{\frac{n \log n}{m}} \right)$ where it is not possible to directly estimate the unknown shift $\bar{\theta}^*$.

## 4.2 Estimating Landmark Columns up to Columnwise Constant Shifts

**Ranking landmark columns** LOE starts by choosing $\ell$ items as landmark items. Upon relabeling the items, we can assume these are the first $\ell$ items. We utilize the ranking results from the previous section to compute (shifted) estimates of the first $\ell$ columns of $D^*$.

Let $D^*_{-i} \in \mathbb{R}^{n-1}$ denote the $i$th column of $D^*$ with the $i$th entry removed (in MATLAB notation $D^*_{-i} := D^*([1{:}i{-}1, i{+}1{:}n], i)$). We identify $[n] \setminus \{i\}$ with $[n-1]$ via the bijection $j \mapsto j - \mathbf{1}\{j > i\}$. Observe that using our triplet oracle $\mathcal{O}$, we can query the $D^*_{-i}$-oracle $\mathcal{O}_i$ which compares items $[n] \setminus \{i\}$ by their distance to $\mathbf{x}_i$. Namely for $\langle j, k \rangle \in \binom{n-1}{2}$:

$$\mathcal{O}_i(\langle j, k \rangle) := \mathcal{O}(\langle i, j + \mathbf{1}\{j \geq i\}, k + \mathbf{1}\{k \geq i\} \rangle). \tag{4.3}$$

By Theorem 2, using $m$ comparisons from $\mathcal{O}_i$ we compute an MLE estimate $R_i$ on Line 6 such that:

$$D^*_{-i} = R_i + s^*_i \mathbf{1}_{n-1} + \epsilon_i, \; i \in [\ell], \tag{4.4}$$

with shift $s^*_i = \frac{1}{n-1}\mathbf{1}^\top_{n-1} D^*_{-i}$. Define the ranking error $\epsilon$ as:

$$\epsilon := \max_{i \in [\ell]} \|\epsilon_i\|_\infty. \tag{4.5}$$

Assuming that Eq. (4.2) in Theorem 2 occurs for each $R_i$, we see that $\epsilon \to 0$ as $m \to \infty$ at a rate $O(1/\sqrt{m})$. From now on we use $\approx$ with respect to this $\epsilon$. The use of this notation is make the exposition and motivation for the algorithm more clear. We will keep track of the suppressed $g$ carefully in our theoretical sample complexity bounds detailed in Appendix A of the supplementary.

## 4.3 Recovering Landmark Column Shifts $s^*$

**Estimating the shifts $s^*$: A motivating strategy** After solving $\ell$ ranking problems and computing estimates $R_i$ for each $i \in [\ell]$, we wish to estimate the unknown shifts $s^*_i$ so that we can approximate the first $\ell$ columns of $D^*$ using the fact that $D^*_{-i} \approx R_i + s^*_i \mathbf{1}$ and $D^*_{i,i} = 0$. As remarked before, we cannot recover such $s^*_i$ purely from ranking information alone.

To circumvent this issue, we incorporate known structural information about the distance matrix to estimate $s^*_i$. Let $E^* := D^*(1{:}\ell, 1{:}\ell)$ and $F^* := D^*(\ell+1{:}n, 1{:}\ell)$ be the first $\ell$ rows and last $n - \ell$ rows of the landmark columns $D^*(1{:}n, 1{:}\ell)$ respectively. Observe that $E^*$ is simply the distance matrix of the $\ell$ landmark objects. Consider the $(n-1) \times \ell$ ranking matrix $R = [R_1, \ldots, R_\ell]$ and let $W$ be its upper $(\ell-1) \times \ell$ submatrix with a diagonal of zeroes inserted (see Fig. 1a). After shifting column $i$ of $R$ by $s^*_i$, column $i$ of $W$ is shifted by $s^*_i$ with the diagonal unchanged. The resulting shift of $W$ is equivalent to adding $J \cdot \mathrm{diag}(s^*)$, so for shorthand we will define:

$$\mathrm{shift}(X, y) := X + J \cdot \mathrm{diag}(y).$$

By definition of $R$, we have $\mathrm{shift}(W, s^*) \approx E^* \in \mathbb{EDM}^\ell$ (see Fig. 1b). This motivates an initial strategy for estimating $s^*_i$: choose $\widehat{s}$ such that $\mathrm{shift}(W, s)$ is approximately a Euclidean distance matrix. Concretely, we choose:

$$\widehat{s} \in \arg\min_{s \in \mathbb{R}^\ell} \mathrm{dist}(\mathrm{shift}(W, s), \mathbb{EDM}^\ell). \tag{4.6}$$

However, it is not obvious how to solve this optimization problem to compute $\widehat{s}$ or if $\widehat{s} \approx s^*$. We address these issues by modifying this motivating strategy in the next section.

**Estimating the shifts $s^*$ using properties of EDMs** Let $\sigma_{E^*} := \frac{1}{\ell(\ell-1)}\langle E^*, J \rangle$ be the average of the non-diagonal entries of $E^*$. Consider the orthogonal decomposition of $E^*$:

$$E^* = E^*_c - \sigma_{E^*} J$$

where the centered distance matrix $E^*_c := E^* - \sigma_{E^*} J$ is the projection of $E^*$ onto $J^\perp$ and consequently lies in the linear subspace $\mathcal{V} := \mathbb{S}^\ell \cap J^\perp$. Letting $\sigma^* := s^* - \sigma_{E^*} \mathbf{1}$ we see:

$$\mathrm{shift}(W, \sigma^*) = \mathrm{shift}(W, s^*) - \sigma_{E^*} J \approx E^* - \sigma_{E^*} J = E^*_c \in \mathcal{V}$$

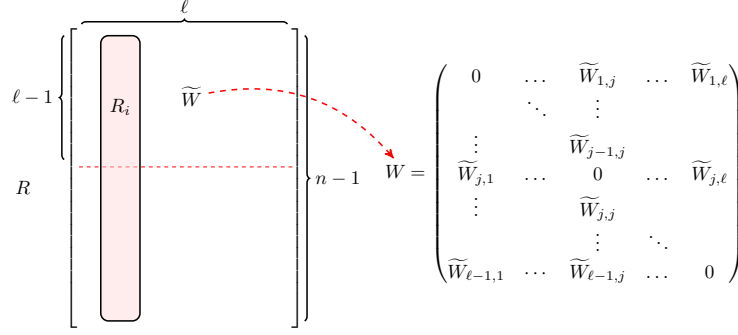

(a) Obtaining $W$ from $R$ (c.f. Line 9 and Line 10). **Left**: The entries of $\widetilde{W}$ are unshifted estimates of the off-diagonal entries $E^*$. **Right**: We add a diagonal of zeroes to $\widetilde{W}$ to match the diagonal of zeroes in $E^*$.

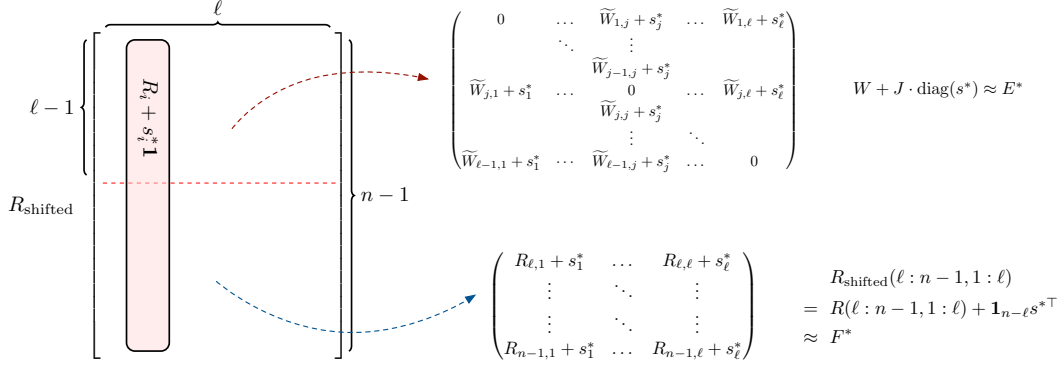

(b) Shifting each $R_i$ by $s_i^*$ to get $E^*$ and $F^*$.

Figure 1: Shifting the rankings $R_i$ to estimate the first $\ell$ columns of $D^*$ (see Algorithm 1).

Since the space $\mathcal{V}$ is more tractable than $\mathbb{EDM}^\ell$ it will turn out to be simpler to estimate $\sigma^*$ than to estimate $s^*$ directly. We will see later that we can in fact estimate $s^*$ using an estimate of $\sigma^*$. In analogy with (4.6) we choose $\widehat{\sigma}$ such that:

$$\widehat{\sigma} \in \arg\min_{s \in \mathbb{R}^\ell} \text{dist}(\text{shift}(W, s), \mathcal{V}) \tag{4.7}$$

This time $\widehat{\sigma}$ is easy to compute. Using basic linear algebra one can verify that the least-squares solution $s_{ls}$ to the system of linear equations:

$$s_i - s_j = W_{ij} - W_{ji}, \forall i < j \in [\ell] \tag{4.8a}$$

$$\sum_{i \in [\ell]} s_i = -\frac{1}{\ell - 1} \sum_{i \neq j \in [\ell]} W_{ij} \tag{4.8b}$$

solves the optimization problem (4.7), so we can take $\widehat{\sigma} := s_{ls}$ (c.f. Line 12). It is not hard to show that $\widehat{\sigma} \approx \sigma^*$ and a proof is given in the appendix (c.f. first half of Appendix A.1 of the supplementary.).

Now it remains to see how to estimate $s^*$ using $\widehat{\sigma}$. To do so we will make use of Theorem 4 of [9] which states that if $\ell \geq d + 3$, then $\sigma_{E^*} = \lambda_2(E_c^*)$ i.e. the second largest eigenvalue of $E_c^*$. If we estimate $\sigma_{E^*}$ by $\widehat{\sigma}_{E^*} := \lambda_2(\mathcal{P}_\mathcal{V}(\text{shift}(W, \widehat{\sigma})))$, then by the theorem $\widehat{\sigma}_{E^*} \approx \sigma_{E^*}$ (c.f. Line 13). Now we can take $\widehat{s} := \widehat{\sigma} + \widehat{\sigma}_{E^*}\mathbf{1}$ (c.f. Line 14) which will satisfy $\widehat{s} \approx s^*$.

After computing $\widehat{s}$, we use it to shift our rankings $R_i$ to estimate columns of the distance matrix. We take $\widehat{E} = \mathcal{P}_{\mathbb{EDM}^\ell}(\text{shift}(W, \widehat{s}))$ and $\widehat{F}$ to be the last $n - \ell$ rows of $R$ with the columns shifted by $\widehat{s}$ (c.f. Line 15 and Line 16). Together, $\widehat{E}$ and $\widehat{F}$ estimate the first $\ell$ columns of $D^*$. We finish by applying LMDS to this estimate (c.f. Line 17).

More generally, the ideas in this section show that if we can estimate the difference of distances $D_{ij}^* - D_{ik}^*$, i.e. estimate the distances up to some constant, then the additional Euclidean distance structure actually allows us to estimate this constant.

# 5 Theoretical Analysis

We now analyze both the sample complexity and computational complexity of LOE.

## 5.1 Sample Complexity

We first present the key lemma which is required to bound the sample complexity of LOE.

**Lemma 3.** *Consider $n$ objects $\mathbf{x}_1, \ldots, \mathbf{x}_n \in \mathbb{R}^d$ with distance matrix $D^* = [D_1^*, \ldots, D_n^*] \in \mathbb{R}^{n \times n}$. Let $\ell = d + 3$ and define $\epsilon$ as in Eq. (4.5). Let $\widehat{E}, \widehat{F}$ be as in Line 15 and Line 16, of LOE (Algorithm 1) respectively. If $E^* = D^*(1 : \ell, 1 : \ell), F^* = D^*(\ell + 1 : n, 1 : \ell)$ then,*

$$\left\| \widehat{E} - E^* \right\| = O(\epsilon \ell^2 \sqrt{\ell}), \left\| \widehat{F} - F^* \right\| = O\left( \epsilon \ell^2 \sqrt{n - \ell} \right).$$

The proof of Lemma 3 is deferred to the appendix. Lemma 3 gives a bound for the propagated error of the landmark columns estimate in terms of the ranking error $\epsilon$. Combined with a perturbation bound for LMDS, we use this result to prove the following sample complexity bound.

**Theorem 4.** *Let $\widehat{\mathbf{X}}$ be the output of LOE with $\ell = d + 3$ landmarks and let $\mathbf{X} \in \mathbb{R}^{d \times n}$ be the true embedding. Then $\Omega(d^8 n \log n)$ triplets queried in LOE is sufficient to recover the embedding i.e. with probability at least $1 - O(dn^{-5})$:*

$$\frac{1}{\sqrt{nd}} d(\widehat{\mathbf{X}}, \mathbf{X}) = O(1).$$

Although the dependence on $d$ in our given bound is high, we believe it can be significantly improved. Nonetheless, the rate we obtain is still polynomial in $d$ and $O(n \log n)$, and most importantly proves the statistical consistency of our method. For most ordinal embedding problems, $d$ is small (often 2 or 3), so there is not a drastic loss in statistical efficiency. Moreover, we are concerned primarily with the large-data regime where $n$ and $m$ are very large and computation is the primary bottle-neck. In this setting, LOE arrives at a reasonable embedding much more quickly than other ordinal embedding methods. This can be seen in Figure 2 and is supported by computational complexity analysis in the following section. LOE can be used to warm-start other more accurate methods to achieve a more balanced trade-off as demonstrated empirically in Figure 3.

We choose the number of landmarks $\ell = d + 3$ since these are the fewest landmarks for which Theorem 4 from [9] applies. An interesting direction for further work would be to refine the theoretical analysis to better understand the dependence of the error on the number of samples and landmarks. Increasing the number of landmarks decreases the accuracy of the ranking of each column since $m/\ell$ decreases, but increases the stability of the LMDS procedure. A quantitative understanding of this trade-off would be useful for choosing an appropriate $\ell$.

## 5.2 Computational Complexity

Computing the regularized MLE for a ranking problem amounts to solving a regularized logistic regression problem. Since the loss function for this problem is strongly convex, the optimization can be done via gradient descent in time $O(C \log \frac{1}{\epsilon})$ where $C$ is the cost of a computing a single gradient and $\epsilon$ is the desired optimization error. If $m$ total triplets are used so that each ranking uses $m/\ell$ triplets, then $C = O(m/\ell + n)$. Since $\ell = O(d)$, solving all $\ell$ ranking problems (c.f. Line 6) with error $\epsilon$ takes time $O((m + nd) \log \frac{1}{\epsilon})$.

Let us consider the complexity of the steps following the ranking problems. A series of $O(nd + d^3)$ operations are performed in order to compute $\widehat{E}$ and $\widehat{F}$ (c.f. Line 16). The LMDS procedure then takes $O(d^3 + nd^2)$ time. Thus overall, these steps take $O(d^3 + nd^2)$ time.

If we treat $d$ and $\log(1/\epsilon)$ as constants, we see that LOE takes time linear in $n$ and $m$. Other ordinal embedding methods, such as STE, GNMDS, and CKL require gradient or projection operations that take at least $\Omega(n^2 + m)$ time and need to be done multiple times.

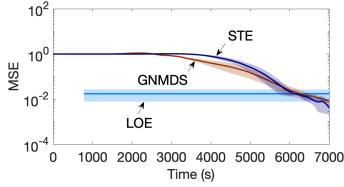

(a) $(n, d, c) = (10^5, 2, 200)$

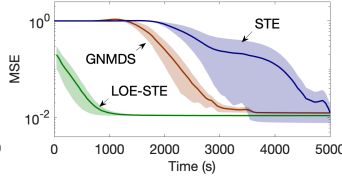

(a) $(n, d, c) = (10^5, 2, 50)$

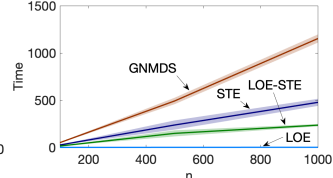

(a) Time to completion vs $n$

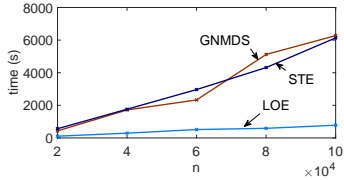

(b) Time to LOE error vs $n$

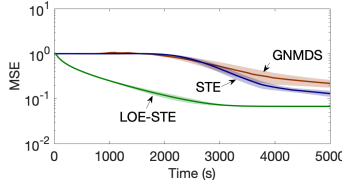

(b) $(n, d, c) = (2 \times 10^4, 10, 200)$

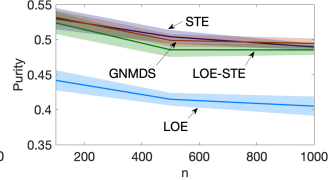

(b) Purity vs $n$

Figure 2: Scalability      Figure 3: Warm-start      Figure 4: MNIST

## 6 Experiments

### 6.1 Synthetic Experiments

We tested the performance of LOE on synthetic datasets orders of magnitude larger than any dataset in previous work. The points of the latent embedding were generated by a normal distribution: $x_i \sim \mathcal{N}(0, \frac{1}{\sqrt{2d}} I_d)$ for $1 \leq i \leq n$. Triplet comparisons were made by a noisy BTL oracle. The total number of triplet queries $m$ for embedding $n$ items was set to be $cn \log n$ for various values of $c$. To evaluate performance, we measured the Procrustes error (3.2) with respect to the ground truth embedding. We compared the performance of LOE with the non-convex versions of STE and GNMDS. For fairness, we did not compare against methods which assume different noise models. We did not compare against other versions of STE and GNMDS since, as demonstrated in the appendix, they are much less computationally efficient than their non-convex versions.

**Scalability with respect to $n$** In this series of experiments, the number of items $n$ ranged over $[20, 40, 60, 80, 100] \times 10^3$. The embedding dimension $d = 2$ was fixed and $c$ was set to 200. For each $n$, we ran 3 trials. In Figure 2a we plot time versus error for $n = 10^5$. In Figure 2b, for each $n$ we plotted the time for LOE to finish and the time for each baseline to achieve the error of LOE. See appendix for remaining plots.

From Figure 2a we see that LOE reaches a solution very quickly. The solution is also fairly accurate and it takes the baseline methods around 6 times as long to achieve a similar accuracy. In Figure 2b we observe that the running time of LOE grows at a modest linear rate. In fact, we were able to compute embeddings of $n = 10^6$ items in reasonable time using LOE, but were unable to compare with the baselines since we did not have enough memory. The additional running time for the baseline methods to achieve the same accuracy however grows at a significant linear rate and we see that for large $n$, LOE is orders of magnitude faster.

**Warm start** In these experiments we used LOE to warm start STE. We refer to this warm-start method as LOE-STE. To obtain the warm start, LOE-STE first uses $\epsilon m$ triplets to obtain a solution using LOE. This solution is then used to initialize STE, which uses the remaining $(1 - \epsilon)m$ triplets to reach a final solution. Since the warm start triplets are not drawn from a uniform distribution on $\mathcal{T}$, they are not used for STE which requires uniformly sampled triplets. We chose $\epsilon$ to be small enough so that the final solution of LOE-STE is not much worse than a solution obtained by STE using $m$ samples, but large enough so that the initial solution from LOE is close enough to the final solution for there to be significant computational savings.

For $d = 2$ we set $n = 10^5$, $c = 50$, and $\epsilon = 0.3$. For $d = 10$ we set $n = 2 \times 10^4$, $c = 200$, and $\epsilon = 0.2$. For each setting, we ran 3 trials. The results are shown in Figure 3. We see that LOE-STE is able to reach lower error much more rapidly than the baselines and yields final solutions that are competitive with the slower baselines. This demonstrates the utility of LOE-STE in settings where the number of triplets is not too large and accurate solutions are desired.

## 6.2 MNIST Dataset

To evaluate our approach on less synthetic data, we followed the experiment conducted in [14] on the MNIST data set. For $n = 100, 500$, and $1000$, we chose $n$ MNIST digits uniformly at random and generated $200n \log n$ triplets comparisons drawn uniformly at random, based on the Euclidean distances between the digits, with each comparison being incorrect independently with probability $ep = 0.15$. We then generated an ordinal embedding with $d = 5$ and computed a $k$-means clustering of the embedding. To evaluate the embedding, we measure the purity of the clustering, defined as $\text{purity}(\Omega, \mathcal{C}) = \frac{1}{n} \sum_k \max_j |\omega_k \cap c_j|$ where $\Omega = \{\omega_1, \omega_2, \ldots, \omega_k\}$ are the clusters and $\mathcal{C} = \{c_1, c_2, \ldots, c_j\}$ are the classes. The higher the purity, the better the embedding. The correct number of clusters was always provided as input. We set the number of replicates in $k$-means to 5 and the maximum number of iterations to 100. For LOE-STE we set $\epsilon = 0.5$. The results are shown in Figure 4. Even in a non-synthetic setting with a misspecified noise model, we observe that LOE-STE achieves faster running times than vanilla STE, with only a slight loss in embedding quality.

## 6.3 Food Dataset

To evaluate our method on a real data set and qualitatively assess embeddings, we used the food relative similarity dataset from [23] to compute two-dimensional embeddings of food images. The embedding methods we considered were STE and LOE-STE. For LOE-STE, the warm start solution was computed with $\ell = 25$ landmarks, using all available landmark comparisons. STE was then ran using the entire dataset. The cold-start STE method used the entire dataset as well. For each method, we repeatedly computed an embedding 30 times and recorded the time taken. We observed that the warm start solution always converged to the same solution as the cold start (as can be seen in the appendix) suggesting that LOE warm-starts do not provide poor initializations for STE. On average, STE took $9.8770 \pm 0.2566$ seconds and LOE-STE took $8.0432 \pm 0.1227$, which is a 22% speedup. Note however that this experiment is not an ideal setting for LOE-STE since $n$ is small and the data set consists of triplets sampled uniformly, resulting in very few usable triplets for LOE.

## 7 Conclusion

We proposed a novel ordinal embedding procedure which avoids a direct optimization approach and instead leverages the low-dimensionality of the embedding to learn a low-rank factorization. This leads to a multi-stage approach in which each stage is computationally efficient, depending linearly on $n$, but results in a loss of statistical efficiency. However, through experimental results we show that this method can still be advantageous in settings where one may wish to sacrifice a small amount of accuracy for significant computational savings. This can be the case if either (i) data is highly abundant so that the accuracy of LOE is comparable to other expensive methods, or (ii) data is less abundant and LOE is used to warm-start more accurate methods. Furthermore, we showed that LOE is guaranteed to recover the embedding and gave sample complexity rates for learning the embedding. Understanding these rates more carefully by more precisely understanding the effects of varying the number of landmarks may be interesting since more landmarks leads to better stability, but at the cost of less triplets per column. Additionally, extending work on active ranking into our framework may provide a useful method for active ordinal embedding. Applying our method to massive real world data-sets such as those arising in NLP or networks may provide interesting insights into these dataset for which previous ordinal embedding methods would be infeasible.

**Acknowledgements**   Nikhil Ghosh was supported in part by a Caltech Summer Undergraduate Research Fellowship. Yuxin Chen was supported in part by a Swiss NSF Early Mobility Postdoctoral Fellowship. This work was also supported in part by gifts from PIMCO and Bloomberg.

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
