[Supplementary Material · ghosh19-loe-cameraready-long.pdf]

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

# A Proof of Theorem 4

In this section, we provide a proof of the sample complexity bound given in Theorem 4.

## A.1 Proof of Lemma 3

*Proof of Lemma 3.* We start by analyzing the error $\|\widehat{\sigma} - \sigma^*\|$. Fix an ordering of $\binom{\ell}{2}$ corresponding to the order of the equations in (4.8a). Define $\mathbb{S}_h^\ell$ to be the subspace of matrices in $\mathbb{S}^\ell$ with zero diagonal. Let $\mathcal{B} : \mathbb{S}_h^\ell \to \mathbb{R}^{\binom{\ell}{2}+1}$ be the linear map defined as

$$\mathcal{B}(X) = \left( (X_{ij} - X_{ji})_{\langle i,j \rangle \in [\frac{\ell}{2}]}, -\frac{1}{\ell-1}\sum_{i \neq j} X_{ij} \right)$$

Let $A \in \mathbb{R}^{(\binom{\ell}{2}+1) \times \ell}$ be the matrix of coefficients in the linear system Eq. (4.8). Then this system is given by the equation $As = \mathcal{B}(W)$. Define the linear map $\mathcal{L}(s) := \text{shift}(W, s)$.

$$\|\mathcal{B}(\mathcal{L}(s))\| = \|\mathcal{B}(W + J \cdot \text{diag}(s))\| = \|\mathcal{B}(W) + \mathcal{B}(J \cdot \text{diag}(s))\| = \|\mathcal{B}(W) - As\|$$

is the error of $s$. Let $\epsilon$ be the ranking error from Eq. (4.5). Then

$$\begin{aligned}
\|\mathcal{B}(\mathcal{L}(\sigma^*))\| = \|\mathcal{B}(E_c^*) - \mathcal{B}(\mathcal{L}(\sigma^*))\| &&\quad \mathcal{B}(E_c^*) = \mathbf{0} \\
\leq \|\mathcal{B}\|_{op}\|\mathcal{L}(\sigma^*) - E_c^*\| && \\
\leq \|\mathcal{B}\|_{op}\epsilon\sqrt{\ell(\ell-1)} &&\quad \|\mathcal{L}(\sigma^*) - E_c^*\| \leq \epsilon\sqrt{\ell(\ell-1)}.
\end{aligned}$$

Therefore if $\widehat{\sigma} = A^\dagger \mathcal{B}(W)$ is the least-squares solution to (4.8), $\|\mathcal{B}(\mathcal{L}(\widehat{\sigma}))\| \leq \|\mathcal{B}(\mathcal{L}(\sigma^*))\|$ and

$$\begin{aligned}
\|A\widehat{\sigma} - A\sigma^*\| &= \|\mathcal{B}(\mathcal{L}(\widehat{\sigma})) - \mathcal{B}(\mathcal{L}(\sigma^*))\| \\
&\leq 2\|\mathcal{B}(\mathcal{L}(\sigma^*))\| \\
&\leq 2\|\mathcal{B}\|_{op}\epsilon\sqrt{\ell(\ell-1)}
\end{aligned}$$

Since $A$ is injective, $\|\widehat{\sigma} - \sigma^*\| \leq \|A^\dagger\|_{op}\|A(\widehat{\sigma} - \sigma^*)\| \leq 2\|A^\dagger\|_{op}\|\mathcal{B}\|_{op}\epsilon\sqrt{\ell(\ell-1)} = O(\epsilon\ell)$

Therefore, if $\widehat{E}_c := \mathcal{P}_\mathcal{V}(\mathcal{L}(\widehat{\sigma}))$

$$\begin{aligned}
|\sigma_{E^*} - \widehat{\sigma}_{E^*}| = |\lambda_2(E_c^*) - \lambda_2(\widehat{E}_c)| &\leq \left\| E_c^* - \widehat{E}_c \right\| \\
&\leq \|E_c^* - \mathcal{L}(\sigma^*)\| + \|\mathcal{L}(\sigma^*) - \mathcal{L}(\widehat{\sigma})\| \\
&= \epsilon\sqrt{\ell(\ell-1)} + \|J \cdot \text{diag}(\sigma^* - \widehat{\sigma})\| \\
&\leq \epsilon\sqrt{\ell(\ell-1)} + \sqrt{\ell-1}O(\epsilon\ell) \\
&= O(\epsilon\ell\sqrt{\ell})
\end{aligned}$$

and so taking $\widehat{s}$ as in Line 14 we see that

$$\begin{aligned}
\|s^* - \widehat{s}\| &\leq \|\sigma^* - \widehat{\sigma}\| + |\sigma_{E^*} - \widehat{\sigma}_{E^*}|\|\mathbf{1}_\ell\| \\
&\leq O(\epsilon\ell) + \sqrt{\ell}O(\epsilon\ell\sqrt{\ell}) = O(\epsilon\ell^2)
\end{aligned}$$

Taking $\widehat{E}$ and $\widehat{F}$ as in Eq. (15), Eq. (16) respectively we get that

$$\begin{aligned}
\left\| E^* - \widehat{E} \right\| &\leq \|E^* - \mathcal{L}(s^*)\| + \|\mathcal{L}(s^*) - \mathcal{L}(\widehat{s})\| \\
&\leq O(\epsilon\ell) + O(\epsilon\ell^2\sqrt{\ell}) = O(\epsilon\ell^2\sqrt{\ell}) \tag{A.1}
\end{aligned}$$

and if $R_b = R(\ell+1 : n, 1 : \ell)$ then

$$\begin{aligned}
\left\| F^* - \widehat{F} \right\| &\leq \left\| F^* - (R_b + \mathbf{1}_{n-\ell}(s^*)^\top) \right\| + \left\| (R_b + \mathbf{1}_{n-\ell}(s^*)^\top) - (R_b + \mathbf{1}_{n-\ell}(\widehat{s})^\top) \right\| \\
&\leq O(\epsilon\sqrt{\ell(n-\ell)}) + \left\| \mathbf{1}_{n-\ell}(s^* - \widehat{s})^\top \right\| \\
&\leq O(\epsilon\sqrt{\ell(n-\ell)}) + \sqrt{n-\ell}O(\epsilon\ell^2) \\
&= O(\epsilon\ell^2\sqrt{n-\ell}) \tag{A.2}
\end{aligned}$$

which completes the proof. $\qquad\square$

## A.2 Proof of Theorem 4

We now give the proof of Theorem 4. In this section we let $\|\cdot\|$ denote the operator norm and $\|\cdot\|_F$ the Frobenius norm. We use the perturbation bounds from [?] to obtain error bounds for the embedding $\widehat{\mathbf{X}}$ recovered using LMDS with inputs $\widehat{E}, \widehat{F}$.

**Theorem 5** (Theorem 2 from [?]). *Consider a centered and full-rank configuration $Y \in \mathbb{R}^{\ell \times d}$ with distance matrix $E^*$. Let $\widehat{E}$ denote another distance matrix and set $\delta^2 = \frac{1}{2}\left\|H(\widehat{E} - E^*)H\right\|_F$. If it holds that $\left\|Y^\dagger\right\|\delta \leq \frac{1}{\sqrt{2}}(\|Y\|\|Y^\dagger\|)^{-1/2}$, then classical MDS with input distance matrix $\widehat{E}$ and dimension $d$ returns a centered configuration $Z \in \mathbb{R}^{\ell \times d}$ satisfying*

$$d(Z, Y) \leq (\|Y\|\|Y^\dagger\| + 2)\|Y^\dagger\|\delta^2$$

*where $H = I_\ell - \frac{1}{\ell}\mathbf{1}_\ell \mathbf{1}_\ell^\top$.*

**Theorem 6** (Theorem 3 from [?]). *Consider a centered configuration $Y \in \mathbb{R}^{\ell \times d}$ that spans the whole space, and for a given configuration $U \in \mathbb{R}^{(n-\ell) \times d}$ let $F^*$ denote the matrix of distances between $U$ and $Y$. Let $Z \in \mathbb{R}^{\ell \times d}$ be another centered configuration that spans the whole space, and let $\widehat{F}$ be an arbitrary matrix of distances. Then LMDS with inputs $Z$ and $\widehat{F}$, returns $\widetilde{U} \in \mathbb{R}^{(n-\ell) \times d}$ satisfying*

$$\left\|\widetilde{U} - U\right\|_F \leq \frac{1}{2}\|Z^\dagger\|\left\|\widehat{F} - F^*\right\|_F + 2\|U\|\|Z^\dagger\|\|Z - Y\|_F$$
$$+ \frac{3}{2}\sqrt{\ell}(\rho_\infty(Y) + \rho_\infty(Z))\|Z^\dagger\|\|Z - Y\|_F + \|Y\|\|U\|\|Z^\dagger - Y^\dagger\|_F$$

*Proof of Theorem 4.* In order to obtain an error bound for the recovered solution $\widehat{X}$, we first need to bound the error of the classical MDS solution which recovers the landmark points. For this we appeal to Theorem 5. For the conditions of the theorem to hold, we need $\delta = O(1)$. Note that $\delta^2 \leq \frac{1}{2}d\left\|\widehat{E} - E^*\right\|_F$, therefore it is sufficient for $\left\|\widehat{E} - E\right\|_F = O(1/d)$. By (A.1) this would mean that $O(d\ell^2\sqrt{\ell}\epsilon) = O(d^{7/2}\epsilon) = O(1)$. From Theorem 2, $\epsilon = O\left(\sqrt{\frac{n \log n}{m}}\right)$ and so $m = \Omega(d^7 n \log n)$ ensures that $\delta = O(1)$.

Let $Z$ be the MDS solution obtained from $\widehat{E}$ and let $Y = [\mathbf{x}_1, \ldots, \mathbf{x}_\ell]$ be the landmark points. If $m/\ell = \Omega(d^7 n \log n)$, then from Theorem 5 $\|Z - Y\|_F = O(1)$. Using the fact that

$$\rho_\infty(Z) \leq \rho_\infty(Y) + \rho_\infty(Z - Y), \|Z^\dagger\| \leq \|Y^\dagger\| + \|Z^\dagger - Y^\dagger\|$$

and the following inequality from [?]

$$\left\|Z^\dagger - Y^\dagger\right\|_F \leq \frac{2\|Y^\dagger\|^2\|Z - Y\|_F}{(1 - \|Y^\dagger\|\|Z - Y\|)_+^2}$$

it follows that $\|Z^\dagger\|, \|Z^\dagger - Y^\dagger\|, \rho_\infty(Z), \rho_\infty(Z - Y)$ are all $O(1)$. The bound in Theorem 6 now reduces to $\left\|\widetilde{U} - U\right\|_F = O\left(\left\|\widehat{F} - F\right\|_F + \sqrt{\ell}\right)$ which by (A.2) is $O(\epsilon\ell^2\sqrt{(n-\ell)} + \sqrt{\ell})$. Therefore

$$\frac{1}{\sqrt{d(n-d)}}\left\|\widetilde{U} - U\right\|_F = O(\epsilon d\sqrt{d}) = O(1)$$

showing that $m/\ell = \Omega(d^7 n \log n)$ triplets per landmark i.e. $m = \Omega(d^8 n \log n)$ total triplets, is sufficient for recovering the embedding.

$\square$

# B Supplemental Experiments

## B.1 Alternative Baselines

In our experiment section, we did not include the results from other complicated versions of STE and GNMDS since they are computationally much less efficient than the non-convex versions considered

in the main paper. Other versions of STE and GNMDS on can consider are STE-Ker, STE-Proj, and GNMDS-Ker. Each of these algorithms maximizes the likelihood of the kernel matrix using projected gradient descent. The methods STE-Ker and GNMDS-Ker project onto the cone of positive semidefinite matrices by zeroing out any negative eigenvalues, a procedure which can be done with an eigenvalue decomposition in $O(n^3)$ time. The method STE-Proj instead projects onto the top $d$ eigenvalues, which takes $O(dn^2)$ time. All three of these methods are extremely computationally inefficient for large $n$. As shown in Figure 5, for even $n = 1000$ it is clear that these methods barely make any progress by the time the non-convex versions converge.

Figure 5: Simulation results: Time vs Performance for $d = 2$, $m = 200n \log n$ with $n = 1000$.

## B.2 Supplemental Computational Efficiency Results

In this section, we display additional results from the scalability experiment in 6.1 that demonstrate the computational efficiency of LOE. Namely we include plots for other values of $n$.

In addition to the Procrustes error (Eq. (3.2)), which is used as the performance measure in the main paper, we also show the triplet prediction error. Let

$$y(\mathbf{x}_i, \mathbf{x}_j, \mathbf{x}_k) = \text{sign}(\|\mathbf{x}_i - \mathbf{x}_k\|^2 - \|\mathbf{x}_i - \mathbf{x}_k\|^2).$$

For an embedding estimate $\widehat{\mathbf{x}}_1, \dots, \widehat{\mathbf{x}}_n$, we define the prediction error as

$$\mathbb{P}\left[y(\widehat{\mathbf{x}}_i, \widehat{\mathbf{x}}_j, \widehat{\mathbf{x}}_k) \neq y(\mathbf{x}_i, \mathbf{x}_j, \mathbf{x}_k)\right]$$

where the triplet $\langle i, j, k \rangle$ is drawn uniformly at random from $\mathcal{T}$. The results are shown in Fig. 6. As expected, we see that LOE recovers the embedding in terms of triplet prediction error as well.

## B.3 Consistency Results

To demonstrate the statistical consistency of LOE, we conducted a set of experiments on synthetic data where $n$ and $d$ are kept fixed, but the number of triplets $m = cn \log n$ varies.

For $n = 100$, $d = 2$ we ranged $c$ over $[10, 50, 100, 150, 200, 300, 400]$. For $n = 100$, $d = 10$, we ranged $c$ over $[100, 200, 600, 1000, 1500, 2000]$. Points were generated from both the normal distribution from 6.1 and also the uniform distribution $x_i \sim \mathcal{U}(0, 1)^d$. For each setting, we averaged over 30 trials. The results are in Figure 7. We see that in each setting with enough triplets the error of LOE goes to 0.

## B.4 Food Dataset Embedding Visualization

In Figure 8 we visualize the embedding results from using LOE-STE and STE on the food dataset mentioned in 6.3.

(a) $n = 20000$

(b) $n = 40000$

(c) $n = 60000$

(d) $n = 80000$

Figure 6: Simulation results: Time vs Performance for $d = 2$, $m = 200n \log n$.

Figure 7: Simulation results: Consistency plots. We fix the number of objects to be $n = 100$. Each scenario corresponds to a different combination of data distribution $\mathbb{D} \in \{$uniform, normal$\}$, and embedding dimension $d \in \{2, 10\}$). Here, we plot the performance of our algorithm as the number of triplets $m = cn \log n$ increases. The plots demonstrate the statistical consistency of our algorithm.

Figure 8: Visualization of the clustering results on the Food dataset: (Top) STE; (Bottom) LOE-STE. As shown in the figures, the embeddings computed by LOE-STE and STE) are identical.