[Reviews · NeurIPS 2019]

Reviewer 1



This is a technically nontrivial paper about ordinal embedding with novel and interesting results and ideas. Unlike most of existing work on ordinal embeddings that focus mainly on investingating the number of comparisons required, this paper aims to reduce the computational complexity of calculating the embedding for large numbers of items. As a consequence it does require a larger number of comparisons, but the gains in computational efficiency seem substantial. The paper is rather dense, and might perhaps benefit from moving some of the techincal description to the appendix and using the resulting space for a more high level overview. (In concrete this could be done by expanding the 1st paragraph of Section 4.) However, even as such the paper reads fairly well. Minor detail: Should there be a reference to some particular part of the appendix on line 144?

Reviewer 2



The techniques are fairly original, the problem is well-motivated and the paper is well-written, thanks to a consistent (though very cumbersome) notation. Apart from some typos (e.g. line 136), I didn't observe major issues in the writeup. I would suggest, however, to give more English descriptions of the mathematical expressions in Section 4, and if possible include some figures (as Figure 1) to explain the algorithm. The result is significant in the sense that I think it will find many applications.

Reviewer 3



Originality: The paper mixes ordinal embedding with Landmark MDS idea, which allow to provide a fast algorithm. This seems new to me and is rather effectice. Quality: The main theoretical result is the Theorem 4 on sample complexity of the method. It states that the algorithm is consistant with high probability given that more than $d^7n\log n$ queries on the ordinal information are done, in the BTL model. Clarity: The paper is very clearly written and insights that generated the idea are well introduced and explained. Significance: The method itself being the most effective given the experiment results, it is certainly significant.

Reviewer 4



This paper studies the problem of recovering n vectors in R^d (up to an isometry by definition of the Procrustes error) based on ordinal feedback. More precisely, the learner observes comparisons between triplets (i,j,k) similar to "j is closer to i than k" sampled from an oracle characterized by a link function. The approach proposed by the authors consist in selecting l landmark items and estimating the corresponding l columns of the distance matrix. This estimation is in 2 steps: i) maximum likelihod estimation under the Bradley-Terry-Luce model (as in ranking via pairwise comparisons) and ii) estimation of the column shifts. Sample complexity bounds are provided. Experiments on both synthetic and real datasets illustrate the efficiency of LOE methods. Pros. - globally well motivated and well written - well connected with related works (e.g. use of LOE as a warm start for STE) - experimental part Cons./Comments a) no discussion on the link function: what happens if f is not the logistic function? does LOE generalize? b) Procrustes error not well introduced: give reference or discuss this choice more. Moreover, theoretical error bounds such as Theorem 4 are not expressed in terms of the Procrustes measure. c) replace [n]\i by [n]\{i} (set difference notation) d) Avoid using the "consistent estimation" notation defined at line 104 as it is not explicit: it hides the dependency on epsilon and g. e) Lack of accuracy of the theoretical bounds because of the use of big O notations: constants are hidden. For example in Theorem 2, what is the influence of the regularization parameter lambda? Or in Theorem 4, what does "with high probability" means? (instead say: "with proba larger than 1-delta" and show the dependency on delta in the bound). In general, instead of writing A=O(B), rather write A <= c B and ideally give the expression of the constant c, or at least give the constants on which c depend (e.g. lambda, etc.). f) Rewrite the bound in Theorem 4 to show the rate of decay with respect to the sample budget m (instead of only using the fact that (m/l)=Omega(d^7nlogn)). And discuss the choice l=d+3 in Theorem 4. g) section 4.3 not well structured. Rather summarize it in lemma(s) and/or proposition(s) and give the underlying intuitions.

[Author Response · NeurIPS 2019]

1. We thank the reviewers for their valuable suggestions. Please find our answers (**A**) for each reviewer (**R**) below.

2. **R1, R2**: *Clarity and technical explanations*

3. **A**: In order to improve the clarity of the technical discussion in section 4, the final version will include: a more
4. detailed high-level overview at the beginning, more English explanations of mathematical expressions (esp. in the last
5. subsection), and another figure similar to Figure 1 illustrating the algorithm (see below).

6. **R3**: *Generalization of the "shift idea" to other preference models*

7. **A**: The "shift idea" relies on estimating the difference of distances $D_{ij} - D_{ik}$. This determines the distances up to
8. some constant shift, so we estimate the proper shifts in order to recover the distances. This idea can be applied to any
9. linear triplet model which says $j$ is farther from $i$ than $k$ with probability $P_{ijk} = F(D_{ij} - D_{ik})$, where $F$ is the CDF
10. of some 0 symmetric random variable. If we can estimate the difference of distances then the rest of the algorithm
11. carries through. The two most common models of this form (see [1]) are the BTL model ($F$ is the logistic CDF) and the
12. Thurstone model ($F$ is the normal CDF). In future work, it may be possible to extend these general landmark techniques
13. to more structured embedding settings such as in [2].

# References

15. [1] Manuela Cattelan. Models for paired comparison data: A review with emphasis on dependent data. *Statistical*
16. *Science*, pages 412–433, 2012.

17. [2] Shuo Chen, Josh L Moore, Douglas Turnbull, and Thorsten Joachims. Playlist prediction via metric embedding. In
18. *Proceedings of the 18th ACM SIGKDD international conference on Knowledge discovery and data mining*, pages
19. 714–722. ACM, 2012.


[Meta-Review · NeurIPS 2019]

In spite of their presentation that should be clearly improved, the results of this paper have been judged sufficiently sound and novel to recommend acceptance.